EMBO
Molecular Medicine

# OTUB1 triggers lung cancer development by inhibiting RAS monoubiquitination

Maria Francesca Baietti[1,2,‡], Michal Simicek[1,2,†,‡], Layka Abbasi Asbagh[1,2], Enrico Radaelli[1,2],
Sam Lievens[3,4], Jonathan Crowther[1,2], Mikhail Steklov[1,2], Vasily N Aushev[1,2,5], David Martínez García[1,2],
Jan Tavernier[3,4] & Anna A Sablina[1,2,*]

## Abstract

Activation of the RAS oncogenic pathway, frequently ensuing from mutations in RAS genes, is a common event in human cancer. Recent reports demonstrate that reversible ubiquitination of RAS GTPases dramatically affects their activity, suggesting that enzymes involved in regulating RAS ubiquitination may contribute to malignant transformation. Here, we identified the de-ubiquitinase OTUB1 as a negative regulator of RAS mono- and di-ubiquitination. OTUB1 inhibits RAS ubiquitination independently of its catalytic activity resulting in sequestration of RAS on the plasma membrane. OTUB1 promotes RAS activation and tumorigenesis in wild-type RAS cells. An increase of OTUB1 expression is commonly observed in non-small-cell lung carcinomas harboring wild-type KRAS and is associated with increased levels of ERK1/2 phosphorylation, high Ki67 score, and poorer patient survival. Our results strongly indicate that dysregulation of RAS ubiquitination represents an alternative mechanism of RAS activation during lung cancer development.

**Keywords** lung cancer; RAS; reversible ubiquitination
**Subject Categories** Cancer; Respiratory System

## Introduction

The RAS small GTPases (HRAS, NRAS, and KRAS) are essential regulators of diverse eukaryotic cellular processes, such as cell proliferation, cytoskeletal assembly and organization, and intracellular membrane trafficking [for review (Colicelli, 2004)]. The RAS family members function as molecular switches alternating between an inactive GDP-bound and active GTP-bound state. The transition between GDP- and GTP-bound forms is tightly regulated by guanine nucleotide exchange factors (GEF) and GTPase-activating proteins (GAP).

The RAS small GTPases play a major role in the development of human cancer. Oncogenic RAS mutations occur in up to 30% of non-small-cell lung carcinomas (NSCLC), mostly adenocarcinomas (Karnoub & Weinberg, 2008). The intrinsic GTP hydrolysis activity of RAS is the predominant target of most common somatic mutations found in the oncogenic variants of RAS alleles (Pylayeva-Gupta et al, 2011). KRAS-mutant lung adenocarcinomas have higher levels of the MAPK pathway activation than wild-type (wt) KRAS tumors. However, the MAPK cascade is also hyperactivated in a significant proportion of wt KRAS tumors, suggesting that RAS proteins may be frequently activated by alternative mechanisms not yet fully elucidated (Network, 2014).

Beyond oncogenic mutations of RAS, up-regulation of RAS-specific GEFs and functional loss of GAPs also have been shown to contribute to cancer development and progression (Vigil et al, 2010). In addition, several post-translational modifications, such as phosphorylation (Bivona et al, 2006), lipidation (Hancock, 2003), and acetylation (Yang et al, 2013), are known to regulate the functions of RAS GTPases. RAS stability is controlled by the E3 ubiquitin ligases, β-TrCP1, and Nedd4-1, that directly polyubiquitinate RAS proteins triggering degradation (Shukla et al, 2014; Zeng et al, 2014). In addition, we and others have recently demonstrated that RAS family members can undergo reversible mono- and di-ubiquitination (Jura et al, 2006; Xu et al, 2010; Sasaki et al, 2011; Baker et al, 2013a; Simicek et al, 2013). However, how reversible ubiquitination affects RAS activity and its tumorigenic properties remains very much controversial.

Earlier studies reported that reversible ubiquitination restricts the activity of HRAS and NRAS, but not that of KRAS, whereas more recent reports demonstrated that KRAS can also undergo mono- and di-ubiquitination (Jura et al, 2006). Xu et al (2010) demonstrated that di-ubiquitination of HRAS and NRAS by the E3 ubiquitin ligase RABEX5 (RABGEF1) induces their re-localization to the endomembranes,

1   Center for the Biology of Disease, VIB, Leuven, Belgium
2   Center for Human Genetics, KU Leuven, Leuven, Belgium
3   Department of Medical Protein Research, VIB, Leuven, Belgium
4   Department of Biochemistry, Gent University, Gent, Belgium
5   Institute of Carcinogenesis, Blokhin Russian Cancer Research Center, Moscow, Russia
    *Corresponding author. Tel: +32 16330790; Fax: +32 16330145; E-mail: anna.sablina@cme.vib-kuleuven.be
    ‡ These authors equally contributed to this work
    † Present address: PNAC, MRC, LMB, Cambridge, UK

leading to a decrease in RAS activity and downstream signaling. On the other hand, two other groups demonstrated that monoubiquitination of HRAS at Lys117 accelerates intrinsic nucleotide exchange and promotes GTP loading, whereas monoubiquitination of KRAS at Lys147 impaired NF1-mediated GTP hydrolysis (Sasaki *et al*, 2011; Baker *et al*, 2013a,b). Moreover, the KRAS gene fusion with the ubiquitin-conjugating enzyme UBE2L3 has been identified in metastatic prostate cancer. The UBE2L3-KRAS fusion protein is highly ubiquitinated and exhibits transforming activity via specific activation of AKT and p38 MAPK pathways (Wang *et al*, 2011).

Taken together, these studies strongly highlight the importance of reversible ubiquitination of RAS-like GTPases governing downstream signaling. These results also suggest that enzymes involved in RAS ubiquitination may contribute to tumorigenic transformation by modulating RAS activity. In this study, we focused on the identification of specific RAS de-ubiquitinating enzymes (DUBs) and their role in cancer development and progression. We found that OTUB1 up-regulation contributes specifically to the development of wt KRAS lung adenocarcinomas by inhibiting reversible ubiquitination of RAS proteins.

## Results

### OTUB1 controls RAS ubiquitination

To identify DUBs involved in the control of RAS ubiquitination, we utilized a targeted mammalian protein–protein interaction (MAPPIT) screen, a two-hybrid technology for the detection of protein-protein interactions in intact mammalian cells (Lievens *et al*, 2009, 2012). As a proof of concept, we applied the MAPPIT system to examine the interactions between HRAS and its known downstream effectors (Fig EV1). GTPase-deficient HRAS G12V-mutant bait gave rise to the robust MAPPIT signals with each of the tested effector preys (Fig EV1), confirming the feasibility of the MAPPIT approach to identify novel RAS regulators.

A targeted MAPPIT screen, in which the HRAS G12V bait was screened against a library of 55 DUBs, identified four DUBs, USP12,

JOSD2, UCHL5, and OTUB1, as potential interactors of HRAS G12V (Fig 1A). We next assessed whether the candidate DUBs could also interact with KRAS G12V and NRAS Q61K. The MAPPIT assay revealed that, in contrast to other tested DUBs, OTUB1 demonstrated a much higher affinity for both NRAS and KRAS compared to random non-specific baits, MAL and eDHFR (Fig 1B). Altogether, the MAPPIT experiments identified OTUB1 (OTU de-ubiquitinase, ubiquitin aldehyde binding 1), a member of the ovarian tumor domain protease (OTU) family of DUBs (Wang *et al*, 2009; Iglesias-Gato *et al*, 2015), as a putative binding partner of RAS proteins (Fig 1A and B).

Using a set of reciprocal immunoprecipitations, we confirmed that OTUB1 interacted with RAS proteins (Fig 1C–F). We found that OTUB1 formed a complex with either wt NRAS or constitutively active form of NRAS Q61K (Fig 1D). Consistently with this observation, both inactive GDP-bound and active GTP-γ-S-bound forms of NRAS interacted with OTUB1 (Fig 1E), indicating that GTP binding does not significantly affect the interaction between RAS and OTUB1. Furthermore, immunofluorescence analysis revealed that OTUB1 and wt KRAS co-localized at the plasma membrane (Fig 1G). These results strongly indicate that OTUB1 interacts with RAS proteins in a GTP-independent manner, suggesting that OTUB1 is an upstream regulator of RAS GTPases.

We next investigated whether OTUB1 is implicated in the regulation of RAS ubiquitination. Consistently with previous reports (Jura *et al*, 2006; Sasaki *et al*, 2011), we found that all RAS proteins undergo mono- and di-ubiquitination (Fig 2A–D). Suppression of OTUB1 with two different shRNAs resulted in increased levels of NRAS monoubiquitination (Fig 2A), whereas overexpression of wt OTUB1 almost completely abolished ubiquitination of RAS proteins (Fig 2B–D).

OTUB1 has recently emerged as a unique DUB that binds to several classes of E2s, including Ubc13 and UbcH5C, and inhibits ubiquitination independently of its proteolytic activity (Nakada *et al*, 2010; Juang *et al*, 2012; Sato *et al*, 2012; Wiener *et al*, 2012). Therefore, we tested whether the catalytic activity of OTUB1 is essential to promote RAS de-ubiquitination. We found that catalytically inactive OTUB1 C91S-mutant (Edelmann *et al*, 2009) as well as wt OTUB1 dramatically decreased the ubiquitination levels of

▶

**Figure 1. The de-ubiquitinase OTUB1 interacts with the RAS GTPases.**

A   A targeted MAPPIT screen identifies several DUBs as putative RAS interactors. A MAPPIT array containing DUB prey library was screened with HRAS G12V as bait. pSEL (+2L)-HRAS G12V was expressed in HEK293T cells together with the indicated prey. BRAF served as a positive control. Each measurement was done in triplicate. The results are expressed as a mean of normalized luciferase activity (leptin-treated cells *vs* leptin-untreated cells). The overall mean value + 2 s.d. served as a threshold.

B   MAPPIT assay confirms the interaction between OTUB1 and RAS proteins. pSEL(+2L) vectors coding RAS proteins were expressed in HEK293T cells together with the indicated prey. Empty vector and two random baits, MAL and eDHFR, were used as negative controls. REM2 and EFHA1 preys that bind to the bait receptor itself were used to evaluate the expression of the RAS baits. The results are expressed as a mean of normalized luciferase activity ± s.e.m (leptin-treated cells *vs* leptin-untreated cells), *n* = 3.

C   NRAS Q61K mutant co-immunoprecipitates with OTUB1. At 48 h post-transfection with Flag-tagged NRAS Q61K and HA-tagged OTUB1 expression constructs, HA-tagged OTUB1 was immunoprecipitated with anti-HA-agarose followed by immunoblotting using anti-Flag or anti-HA antibodies.

D   OTUB1 interacts with wt NRAS and active NRAS-mutant. Flag-tagged NRAS proteins were immunoprecipitated using anti-Flag (M2) agarose from HEK293T cells overexpressing HA-tagged OTUB1 or empty vector (V).

E   GTP binding does not affect the complex formation between NRAS and OTUB1. Recombinant Flag-tagged NRAS was incubated with lysates derived from HEK293T cells expressing HA-tagged OTUB1 in the excess of GTP-γ-S or GDP, followed by immunoblotting with the indicated antibodies.

F   OTUB1 interacts with wt KRAS. Flag-tagged KRAS was immunoprecipitated using anti-Flag (M2) agarose from HEK293T cells overexpressing HA-tagged OTUB1 or empty vector (V).

G   OTUB1 co-localizes with KRAS at the plasma membrane. At 24 h after co-transfection with GFP-tagged KRAS and HA–tagged OTUB1, HeLa cells were immunostained with anti-HA antibody. The outlined areas are shown at higher magnification at the top of each image. Scale bar, 10 μm.

Data information: (C–F) IP, immunoprecipitates; WCL, whole cell lysate.

Source data are available online for this figure.

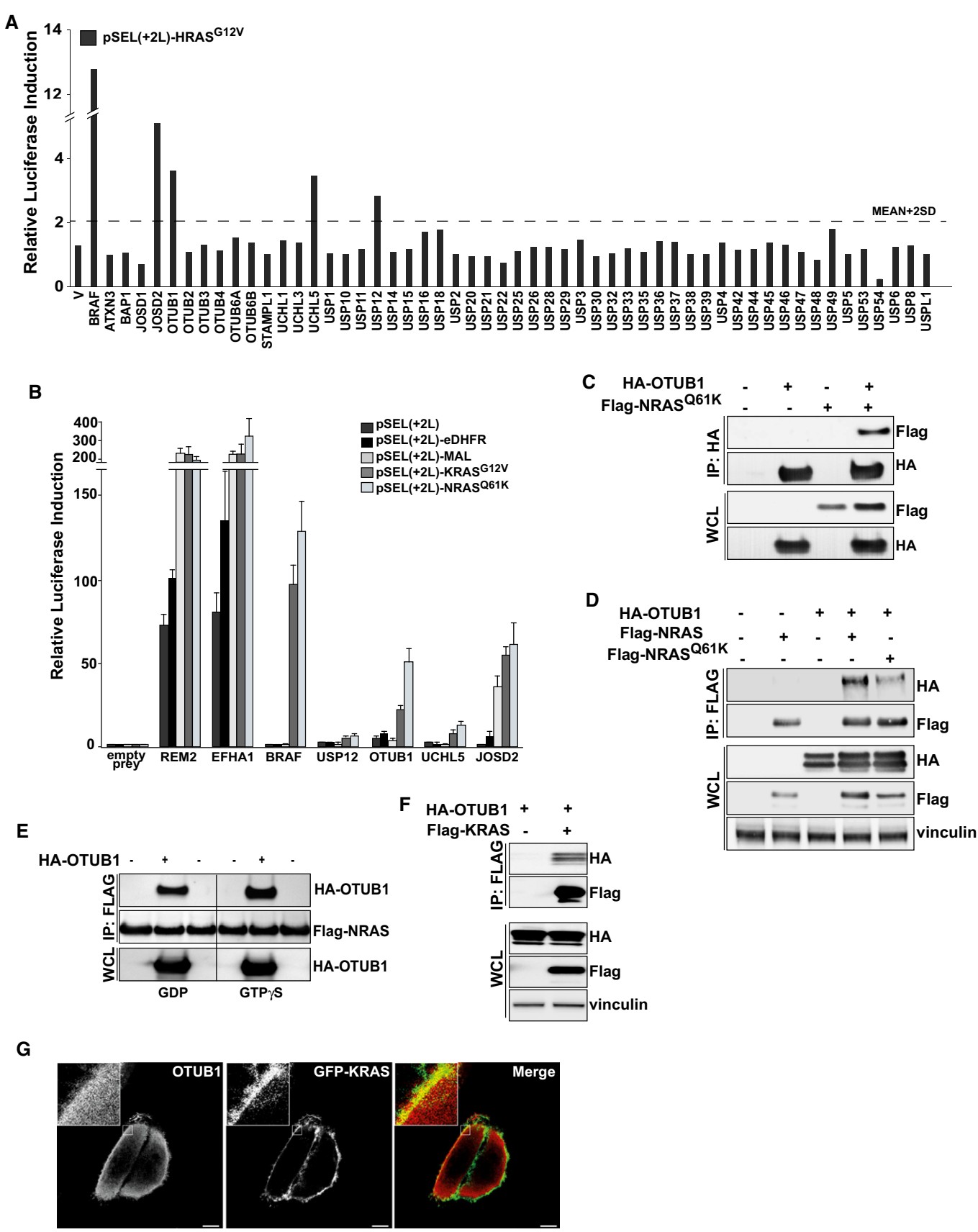

**Figure 1.**

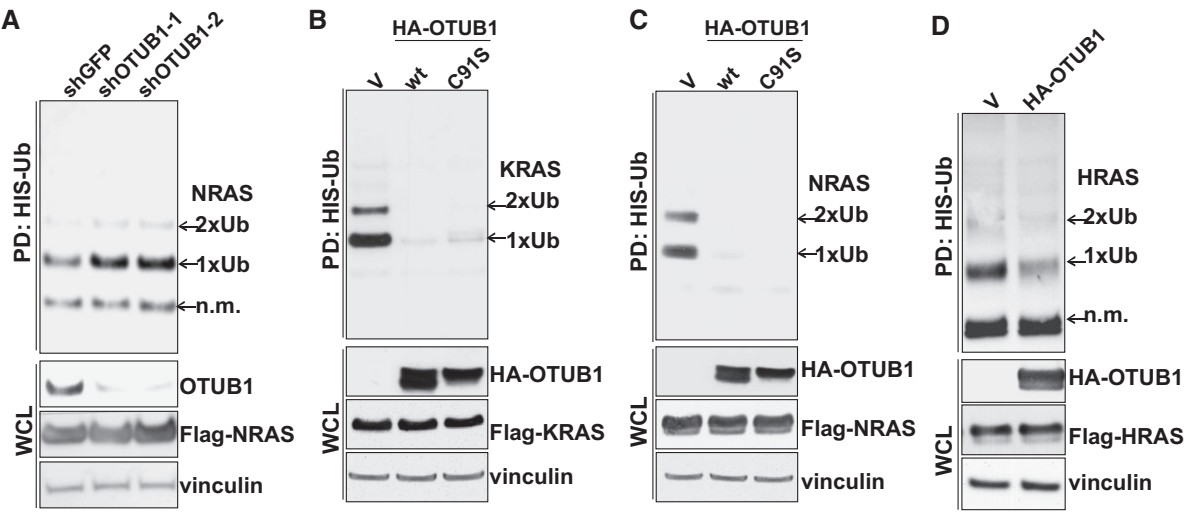

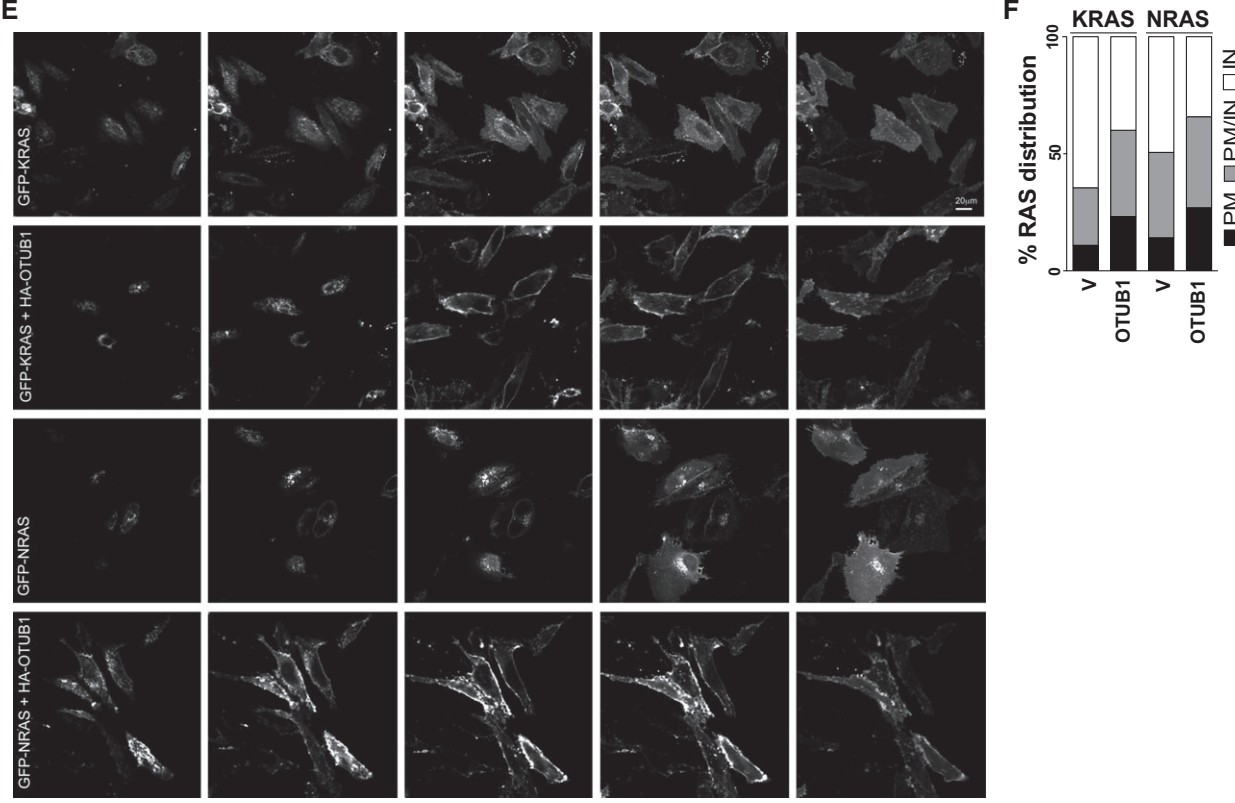

**Figure 2. OTUB1 triggers membrane localization of RAS ubiquitination by inhibiting its ubiquitination.**

A    Suppression of OTUB1 expression increases NRAS mono- and di-ubiquitination. 6×His-tagged ubiquitin and Flag-NRAS were introduced into HEK293T cells expressing shGFP or shRNAs against OTUB1. Ubiquitinated NRAS was purified by $Co^{2+}$ metal affinity chromatography and detected by anti-Flag antibody.

B–D  Catalytic activity of OTUB1 is not required to inhibit RAS ubiquitination. 6×His-tagged ubiquitin and RAS expression constructs were introduced into HEK293T cells expressing wt HA-OTUB1, the catalytically dead mutant HA-OTUB1 C91S, or empty vector (V). Ubiquitinated RAS was purified by $Co^{2+}$ metal affinity chromatography and detected by anti-Flag antibody.

E    OTUB1 induces membrane RAS re-localization. Confocal imaging of HeLa cells expressing the indicated constructs. For each sample, z-stacks obtained by scanning the sample from the apical to the basal layer. Step-size, 2 μm. Scale bar, 20 μm.

F    RAS cellular distribution expressed as percentage of cells with specific RAS localization. For quantification of RAS localization, cells were randomly imaged using IN Cell Analyzer. RAS localization (> 200 cells) was scored as intracellular and diffused (IN), mostly at the plasma membrane (PM), or both intracellular and plasma membrane (PM/IN). *P*-value = 0.0005 as determined by chi-squared test, *n* = 3. Representative images of HeLa cells expressing GFP-tagged KRAS are shown in Appendix Fig S1. pull-down, PD. whole cell lysate, WCL. not modified, n.m.

Source data are available online for this figure.

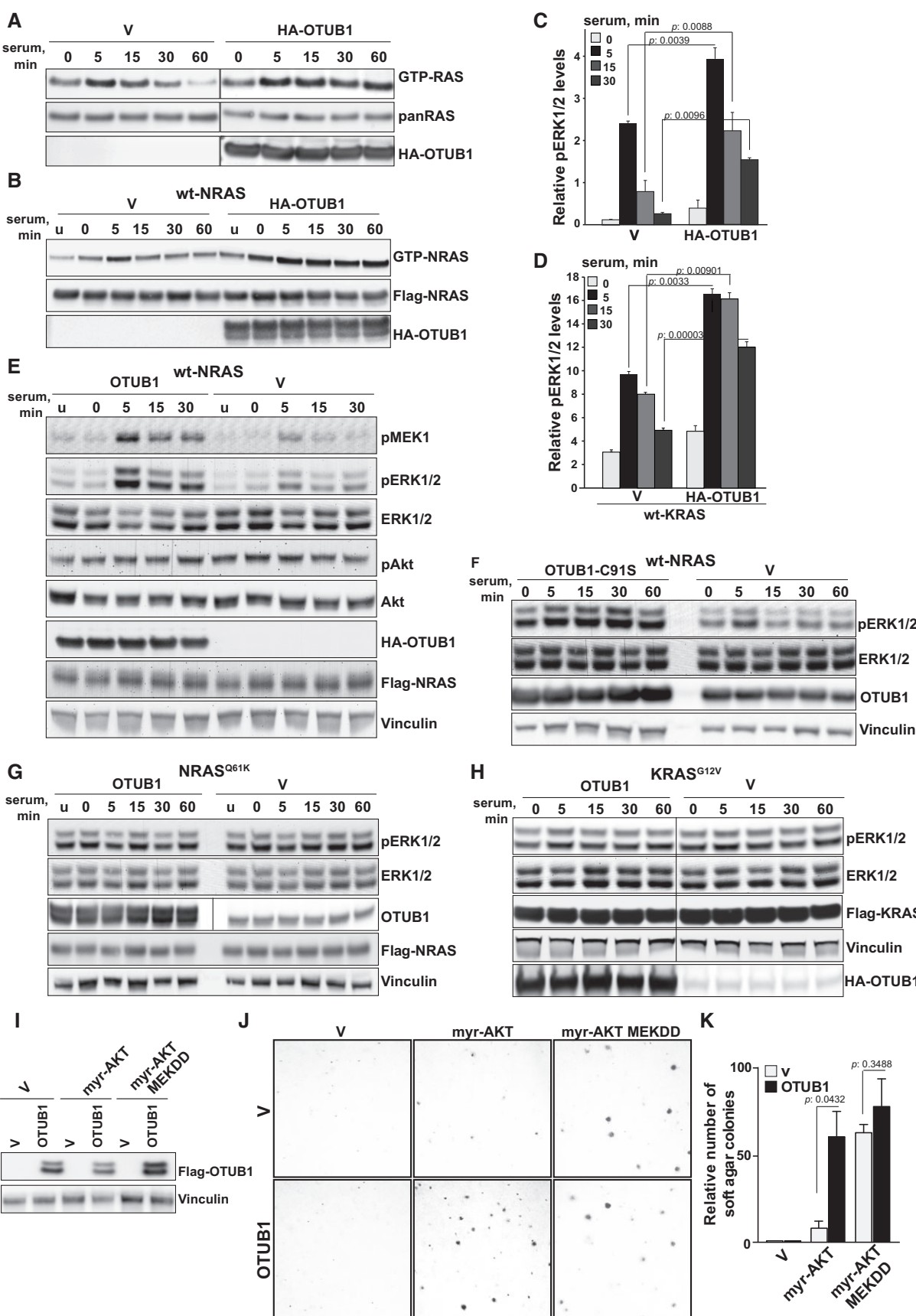

**Figure 3.**

◄

**Figure 3.   OTUB1 increases RAS activity and enhances MAPK activation in wt RAS cells.**

A, B    OTUB1 overexpression promotes serum-induced activation of endogenous wt RAS (A) or wt NRAS (B). GTP-bound RAS was pulled down from HEK293T cells expressing HA-tagged OTUB1 or empty vector (V) using recombinant RAF1 RBD conjugated to agarose beads. Input was controlled by immunoblotting using anti-panRAS or anti-Flag antibodies.

C, D    OTUB1 enhances serum-induced ERK1/2 phosphorylation in endogenous wt RAS (C) or wt KRAS (D). Levels of phosphorylated ERK1/2 and total ERK1/2 in HEK293T cells expressing HA-tagged OTUB1 or empty vector (V) were analyzed by Meso Scale assay. The results are expressed as a mean of pERK1/2 levels relative to total ERK1/2 ± s.e.m. $n$ = 2. *P*-values were determined by two-sided *t*-Test.

E, F    Overexpression of wt OTUB1 (E) or the catalytically dead mutant OTUB1 C91S (F) promotes serum-induced MAPK activation in cell expressing wt NRAS. Whole cell lysates were analyzed by immunoblotting using the indicated antibodies.

G, H    Overexpression of OTUB1 has no effect on serum-induced MAPK activation in cell expressing mutant NRAS Q61K or KRAS G12V. Whole cell lysates were analyzed by immunoblotting using antibodies as indicated.

I    Immunoblot of OTUB1 overexpression in immortalized human embryonic kidney epithelial cells (HEK TEST), expressing empty vector (V), myristoylated AKT1 (myr-AKT), MEK1 D218, D222-mutant (MEKDD).

J, K    OTUB1 cooperates with active AKT1 to promote anchorage-independent growth. Representative images of soft-agar colonies formed by HEK TEST cells expressing the indicated constructs. The number of soft agar colonies formed by cells expressing OTUB1 compared to cells expressing an empty vector. Data are presented as mean ± s.e.m. *P*-values were determined by two-sided *t*-Test, $n$ = 2.

Data information: (A–H) Serum-starved HEK293T cells expressing the indicated constructs were stimulated with 10% serum for the indicated time periods.
Source data are available online for this figure.

RAS, suggesting that OTUB1 may affect RAS ubiquitination by inhibiting the E2 ubiquitin-conjugating enzymes (Fig 2B and C). In fact, *in vitro* ubiquitination of RAS was abolished by wt OTUB1, but not by deltaN(1-30) OTUB1-mutant lacking binding to E2 (Fig EV2A). In contrast, incubation of ubiquitinated RAS with wt OTUB1 did not decrease levels of RAS ubiquitination, thus supporting the premise that OTUB1 functions via E2 inhibition independent of its catalytic activity (Fig EV2B).

Since previous reports demonstrated that reversible ubiquitination of RAS promotes its endosomal association (Jura *et al*, 2006), we tested whether OTUB1 affects the subcellular localization of RAS. Consistent with the observation that OTUB1 inhibits RAS ubiquitination, analysis of RAS localization revealed that OTUB1 overexpression augmented the presence of RAS proteins on the plasma membrane (Fig 2E and F; Appendix Fig S1). Hence, by inhibiting RAS ubiquitination, OTUB1 functions to hinder RAS re-localization from the plasma membrane thereby contributing to the spatial control of RAS-dependent cellular responses.

**OTUB1 triggers RAS activity and downstream signaling**

We next analyzed how OTUB1 affects RAS activity and signaling. We found that overexpression of OTUB1 in HEK293T cells led to hyperactivation of wt RAS upon serum stimulation (Fig 3A and B). In concordance with this result, OTUB1 overexpression triggered a significant increase in phospho-ERK1/2 levels at different time points after addition of serum (Fig 3C–E). We observed a similar overactivation of the MAPK pathway, when we overexpressed catalytically inactive OTUB1 C91S-mutant, indicating that catalytic activity of OTUB1 is not necessary to induce the MAPK pathway activation (Fig 3F). In contrast, OTUB1 overexpression did not dramatically affect phosphorylation levels of AKT1 (Appendix Fig S2A and B). The latter observation could be due to OTUB1-mediated inhibition of TRAF6 (Li *et al*, 2010) that plays a crucial role in AKT activation (Yang *et al*, 2009).

On the other hand, when we overexpressed OTUB1 in HEK293T cells expressing constitutively active RAS-mutants, NRAS Q61K or KRAS G12V, we did not observe any significant up-regulation of ERK1/2 phosphorylation, most likely because the MAPK pathway was already optimally active due to the introduction of the active

RAS-mutants (Fig 3G and H). We also did not observe OTUB1-induced hyperactivation of the MAPK kinase pathway when we overexpressed a dominant-negative KRAS S17N-mutant, indicating that the effect of OTUB1 overexpression is RAS dependent (Appendix Fig S2C). Taken together, these data indicate that OTUB1 up-regulation leads to activation of wt RAS signaling.

**OTUB1 triggers cell transformation by inducing the MAPK cascade activation**

Hyperactivation of the MAPK signaling by OTUB1 overexpression suggests that OTUB1 overexpression may promote tumorigenic transformation. Multiple studies have demonstrated that the co-expression of the telomerase catalytic subunit (hTERT), the SV40 Large T (LT) and small t (ST) oncoproteins, and an activated allele of RAS (RAS G12V) renders a wide range of human cells tumorigenic (Zhao *et al*, 2004), while co-activation of the MAPK and PI3K pathways suffices to replace RAS G12V in human cell transformation (Boehm *et al*, 2007). We used immortalized, but non-malignant human embryonic kidney epithelial cells expressing hTERT, LT, and ST (HEK TEST cells) as a model to assess tumorigenic potential of OTUB1 (Boehm *et al*, 2007). Given that OTUB1 overexpression up-regulated the MAPK pathway, but did not affect AKT signaling, we hypothesized that OTUB1 could cooperate with myristoylated (myr) and therefore the constitutively active allele of AKT1 (myr-AKT) to promote cell transformation. In fact, overexpression of OTUB1 together with myr-AKT1 dramatically induced anchorage-independent growth, whereas OTUB1 alone was not sufficient to trigger soft agar colony formation (Fig 3I–K). On the other hand, OTUB1 did not further accelerate anchorage-independent colony formation of HEK TE cells overexpressing both a constitutively active MEK1 D218, D222 allele (MEKDD) and myr-AKT, further confirming that OTUB1 overexpression promotes tumorigenic transformation by inducing the MAPK cascade activation.

**OTUB1 is more frequently up-regulated in wt KRAS non-small-cell lung carcinomas**

Our results suggest that increased OTUB1 expression could be an alternative mechanism of RAS activation superseding that of RAS

**Figure 4.**  *OTUB1* **expression is up-regulated in wt KRAS lung tumors.**

A       Gains of OTUB1 and KRAS mutation are mutually exclusive in lung adenocarcinomas. OncoPrint showing the distribution of KRAS somatic mutations and OTUB1 copy number alterations in TCGA lung adenocarcinomas and squamous cell carcinomas obtained from cBioPortal (Cerami *et al*, 2012; Gao *et al*, 2013). Co-occurrence analysis showing significant mutual exclusivity between KRAS mutation and OTUB1 gain.

B, C    *OTUB1* overexpression in TCGA lung carcinomas is associated with 11q13.1 copy number alteration. Pearson correlation of *OTUB1* copy number (log2 ratio) with *OTUB1* mRNA levels (RNAseq normalized read counts, log2 transformed) was analyzed.

D–F     *OTUB1* expression in TCGA lung adenocarcinoma (LUAD) and lung squamous cell carcinoma (SCC) patients. Patients were stratified according to their *OTUB1* mRNA levels and/or their KRAS status as described in Materials and Methods. Box whisker plots represent *OTUB1* mRNA expression levels in TCGA lung carcinoma patients determined by RNAseq analysis. *P*-values were determined by two-sided *t*-Test. Total number of patients, n.

G       Gain of 11q13.1 locus is an early event in lung adenocarcinoma development. TCGA lung adenocarcinoma patients with diploid or gain of the *OTUB1* locus were stratified according tumor stages (T1–T4). Total numbers of patients, n. Statistical comparison of the sample distributions were compared using Chi-square test.

H       *OTUB1* mRNA overexpression is an early event in lung adenocarcinoma development. TCGA lung adenocarcinoma patients were stratified by tumor stages (T1–T4) and *OTUB1* expression levels as described in Materials and Methods. Total numbers of patients, n. Statistical comparison of the sample distributions were compared using Chi-square test.

I       KRAS mutation is a late event in lung adenocarcinoma progression. TCGA lung adenocarcinoma patients with different KRAS mutation status were stratified by tumor stages (T1–T4). Total numbers of patients, *n*. Statistical comparison of the sample distributions were compared using Chi-square test.

activating mutations. Analysis of The Cancer Genome Atlas (TCGA) revealed that gain of the 11q13.1 locus, where the *OTUB1* gene resides, was commonly observed in both lung adenocarcinomas and lung squamous cell carcinomas (SCC) (Figs 4A and EV3A). Correlation analysis revealed a strong association between copy number variation of 11q13.1 locus and *OTUB1* expression levels, suggesting that *OTUB1* is commonly up-regulated in lung tumors due to gain of the 11q13 locus (Figs 4B and C, and EV3A and B). *OTUB1* mRNA expression was also significantly up-regulated in about 50% of adenocarcinomas and about 80% of SCC compared to normal tissue samples (Fig 4D–F). These observations are further consolidated by the increase of *OTUB1* in a majority of tumorigenic lesions compared to their respective matched normal samples (Fig EV3C and D).

We also observed a higher proportion of wt KRAS lung adenocarcinomas with medium/high levels of OTUB1 expression compared to mutant KRAS tumors (Fig 4D and F). Correlation analysis revealed that increased expression of OTUB1 (co-occurrence log odds ratio: −1.478; *P*-value: 0.014) or a gain of the OTUB1 locus (co-occurrence log odds ratio: −0.796; *P*-value: 0.017) and the mutation status of KRAS were mutually exclusive (Fig 4A), suggesting that OTUB1 overexpression may play a crucial role in tumorigenesis especially in lung adenocarcinomas harboring wt KRAS.

Furthermore, either 11q13.1 gain or moderate OTUB1 overexpression is observed at early stages in lung adenocarcinomas with no increase in frequency in higher tumor stages (Fig 4G and H). In contrast, the frequency of KRAS mutations significantly increased with tumor stage predominantly associated with later stages of adenocarcinoma progression (Fig 4I). Taken together, these results suggest the role of OTUB1 up-regulation in promoting cancer development in wt KRAS lung tumors.

**OTUB1 enhanced tumorigenic growth of lung adenocarcinomas**

Next we assessed the contribution of OTUB1 to tumorigenic transformation of non-small-cell lung carcinoma (NSCLC) cells. To examine whether OTUB1 is essential for growth, we suppressed OTUB1 expression in lung adenocarcinoma cell lines using OTUB1 shRNAs (Fig 5A). Meso Scale analysis revealed that suppression of OTUB1 in A549 cell line led to decreased ERK1/2 phosphorylation upon serum stimulation (Fig 5B). Stable knockdown of OTUB1 also

decreased anchorage-independent growth of several NSCLC cell lines and dramatically suppressed the xenograft growth of A549 cells in immunocompromised mice (Fig 5C–E). Taken together, these data strongly indicate that OTUB1 expression is essential for NSCLC tumor growth.

To further elucidate tumorigenic activity of OTUB1, we generated stable cell lines expressing Flag-tagged OTUB1. In the generated cell lines, we observed approximately 1.5- to 2-fold increase of OTUB1 expression with respect to endogenous protein levels that corresponds to an increase of OTUB1 expression triggered by 11q13.1 gain (Appendix Fig S3). OTUB1 overexpression in the H1993 cell line harboring wt KRAS led to a higher and more sustained activation of ERK1/2 phosphorylation (Fig 6A), whereas the introduction of OTUB1 into KRAS-mutant A549 cells only slightly increased the activity of the MAPK pathway (Fig 6B). These results are concordant with the effect of OTUB1 overexpression on the MAPK cascade activation in HEK293T cells expressing wt RAS or constitutively active RAS-mutants (Fig 3C–H).

We found that overexpression of OTUB1 in wt KRAS cell lines, H838, H1437, H1993, and HOP92, increased their ability to form colonies in soft agar (Fig 6C–E). In contrast, OTUB1 expression did not significantly affect anchorage-independent growth in KRAS-mutant cell lines, H2009, HOP62, and A427. A sole mutant KRAS A549 cell line had increased colony formation in response to OTUB1 overexpression (Fig 6C–E); however, *in vivo* tumor growth of A549 was not affected upon OTUB1 overexpression (Fig 6F). This indicates that even though OTUB1 is essential to maintain the activity of mutant RAS, up-regulation of OTUB1 expression does not further prompt tumorigenic properties of constitutively active RAS-mutants. Notably, OTUB1 expression in wt KRAS H1993 and H1437 cells significantly enhanced xenograft growth (Fig 6G). Taken together, these results suggest that an increase in OTUB1 expression accelerated tumorigenic transformation of wt KRAS NSCLCs.

**OTUB1 up-regulation is associated with increased ERK1/2 activity in a subset of wt KRAS non-small-cell lung carcinomas**

To confirm the contribution of OTUB1 to lung cancer development, we performed immunohistochemistry analysis of a NSCLC tissue array. OTUB1 immunoreactivity was scored as negative/low, medium, and high (Fig 7A). We found that more than 70% of

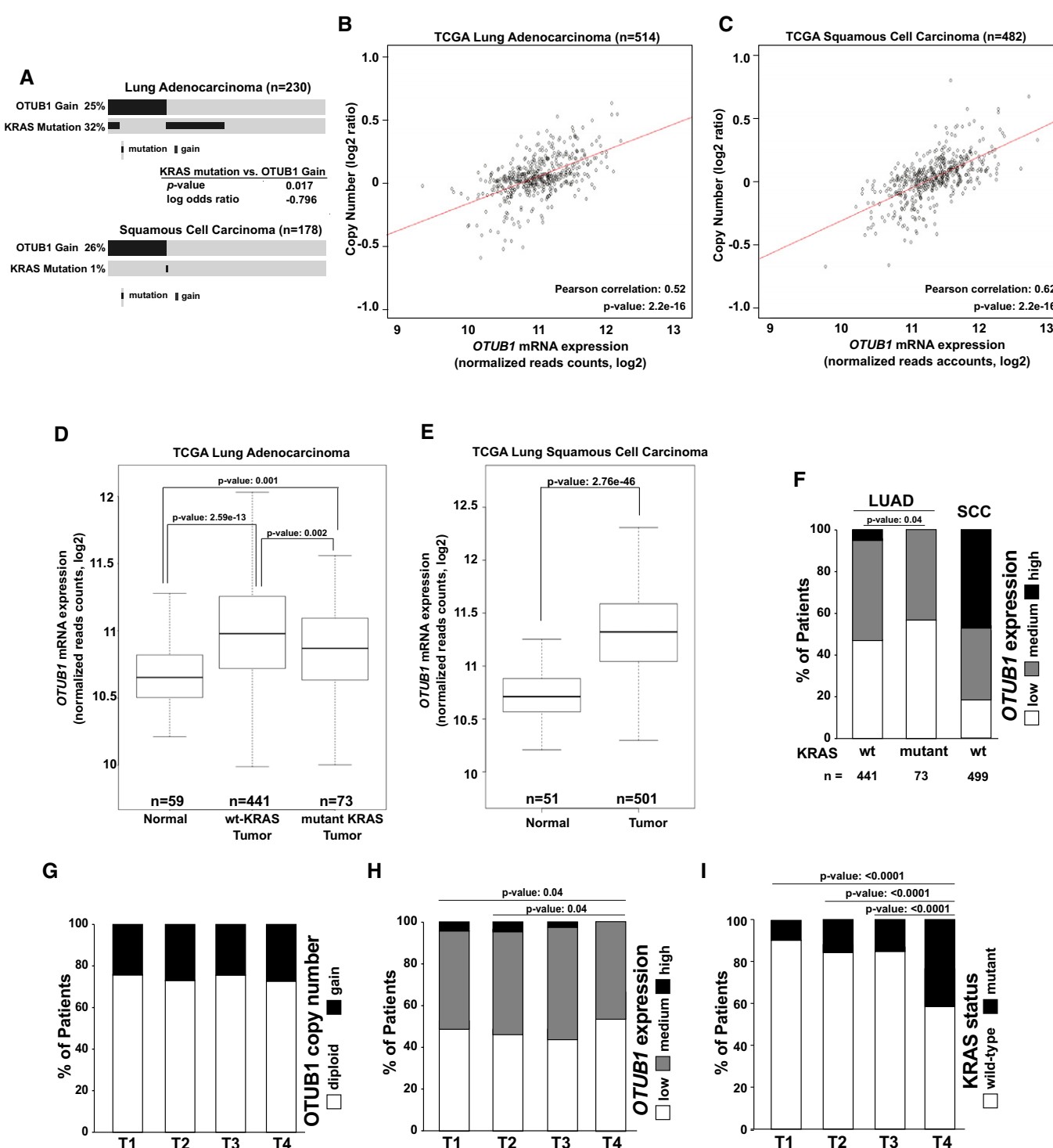

**Figure 4.**

adenocarcinomas and about 25% of squamous cell carcinomas exhibited intermediate or strong cytoplasmic OTUB1 positivity (Figs 7B and EV4A). Consistent with our TCGA data analysis, OTUB1 positivity was already observed in early stages of lung adenocarcinomas with strong immunoreactivity found in stages T2/T3 (Fig 7C). We also stratified the patients according to their

KRAS mutation status. Unfortunately, the low number of mutant KRAS samples ($n = 19$) did not permit statistical evaluation of this subgroup of patients (Fig 7B).

Consistently with our observation that OTUB1 overexpression induces the MAPK cascade activation in cells with wt RAS, we found that higher levels of OTUB1 significantly correlated with

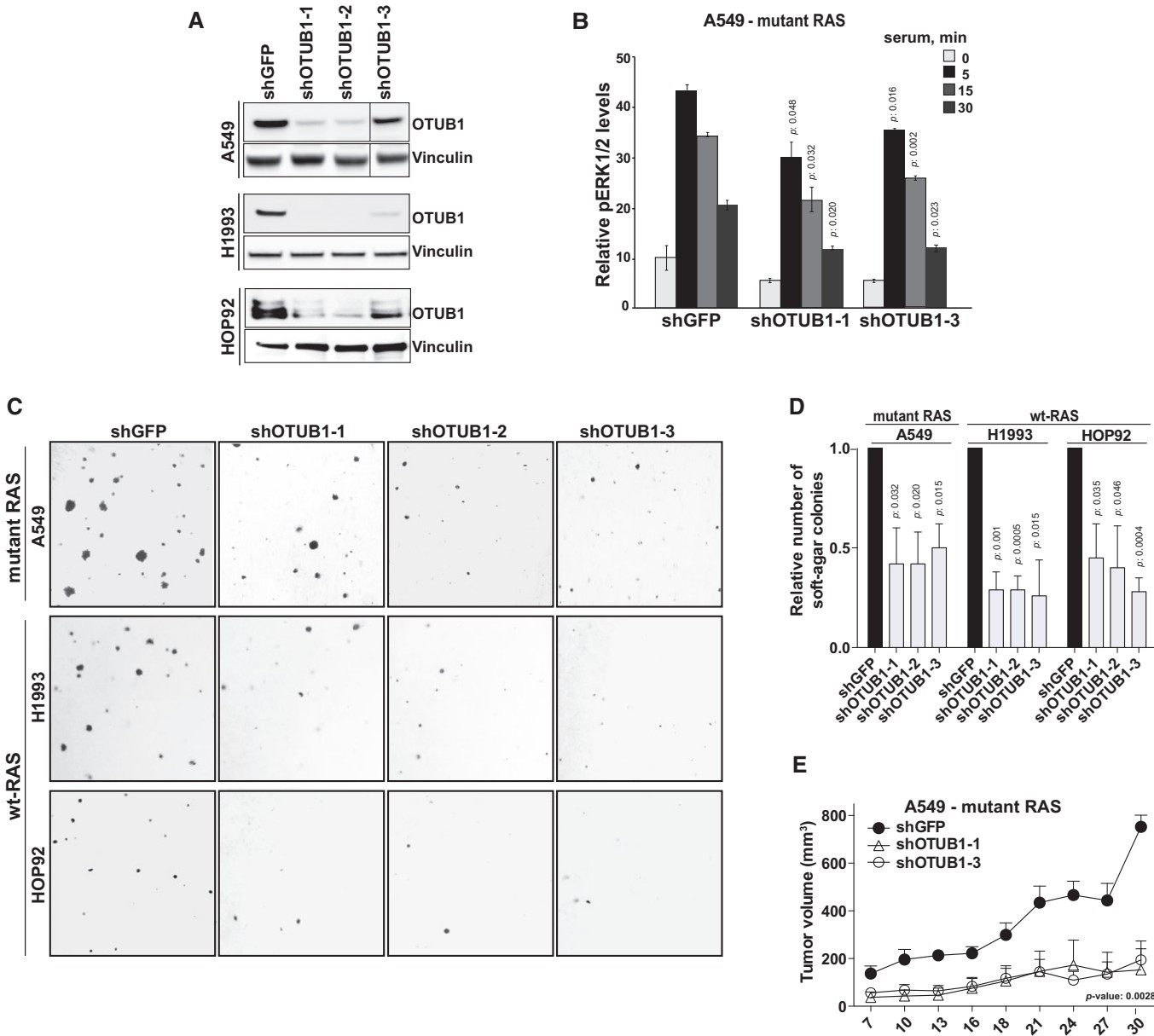

**Figure 5. OTUB1 is essential for tumorigenicity of lung cancer cells.**

A    Suppression of OTUB1 expression in NSCLC cell lines by specific shRNAs against OTUB1 was confirmed by immunoblotting using antibody against OTUB1.

B    OTUB1 suppression decreases serum-induced ERK1/2 phosphorylation. Serum-starved A549 cells expressing shRNAs against OTUB1 or GFP were stimulated with 10% serum for the indicated time periods. Levels of phosphorylated ERK1/2 and total ERK1/2 were analyzed by Meso Scale assay. The results are expressed as a mean of pERK1/2 levels relative to total ERK1/2 ± s.e.m; $P$-values were determined by two-sided $t$-Test, $n = 2$.

C, D    OTUB1 suppression impairs the anchorage-independent growth of lung cancer cell lines. Representative images of soft-agar colonies formed by lung cells expressing shRNAs against OTUB1 or GFP. The number of soft agar colonies formed by cells expressing shOTUB1 compared to cells expressing shGFP. Data are presented as mean ± s.e.m. $P$-values were determined by two-sided $t$-Test, $n = 3$.

E    Xenograft tumor growth of A549 cells expressing shRNAs against GFP or OTUB1 subcutaneously injected into nude mice. Data are presented as mean ± s.e.m. $P$-value was determined by 2-way ANOVA, $n = 4$.

Source data are available online for this figure.

increased levels of ERK1/2 phosphorylation in lung adenocarcinomas harboring wt KRAS (Pearson's coefficient: 0.352; $P$-value: 0.013) (Fig 7D and E), while mutant KRAS tumors exhibited in general higher levels of ERK1/2 phosphorylation (Fig 7D and E).

We also observed that, in wt KRAS tumors, intermediate OTUB1 immunoreactivity has a tendency to display a higher proliferative score, as detected by Ki67 staining (Figs 7F and EV4B). On the other hand, high OTUB1 expression is mostly associated with

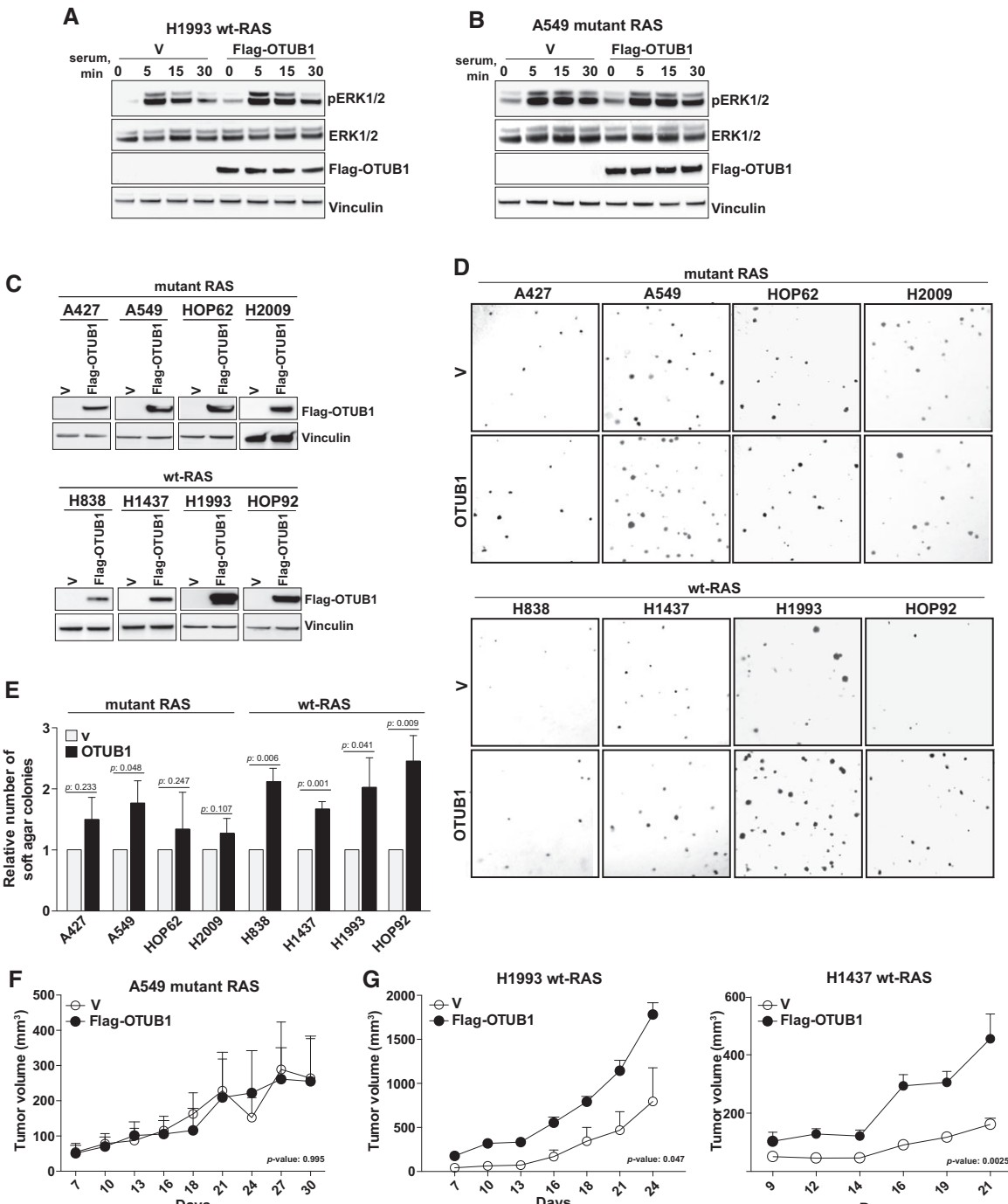

**Figure 6.  OTUB1 promotes tumorigenicity of wt KRAS cells.**

A     OTUB1 overexpression promotes serum-induced MAPK activation in wt KRAS H1993 lung adenocarcinoma cells.

B     OTUB1 overexpression does not affect serum-induced MAPK activation in mutant KRAS A549 lung adenocarcinoma cells.

C     Representative immunoblots of Flag-tagged OTUB1 overexpression in lung adenocarcinoma cell lines.

D, E  OTUB1 overexpression affects anchorage-independent growth of lung adenocarcinoma cell lines. Representative images of soft-agar colonies formed by the indicated cells expressing Flag-OTUB1 or empty vector (V). The number of soft agar colonies formed by OTUB1-expressing cells compared to cells expressing an empty vector. Data are presented as mean ± s.e.m. *P*-values were determined by two-sided *t*-Test, *n* = 3.

F, G  Xenograft tumor growth of A549, H1993 or H1437 cells expressing Flag-OTUB1 or an empty vector (V) subcutaneously injected into nude mice. Data are presented as mean ± s.e.m. *P*-values were determined by 2-way ANOVA, *n* = 4.

Data information: (A, B) Serum-starved lung cells expressing Flag-OTUB1 or an empty vector (V) were stimulated with 10% serum for the indicated time periods. Whole cell lysates were analyzed by immunoblotting using antibodies against pERK1/2, ERK1/2, Flag, and vinculin.

Source data are available online for this figure.

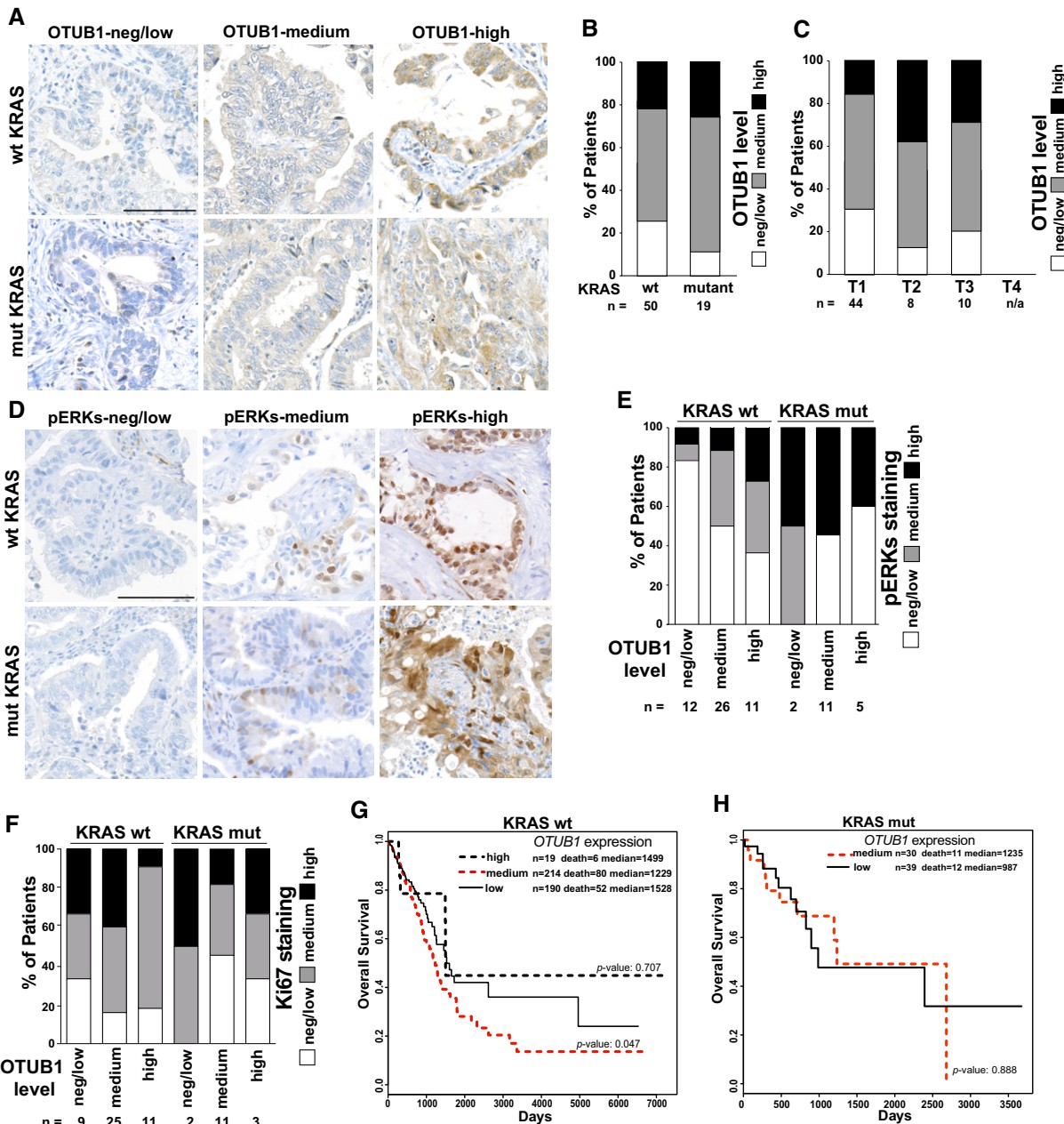

**Figure 7. OTUB1 up-regulation in wt KRAS lung adenocarcinomas correlates with increased MAPK signaling and poorer patient survival.**

A   Representative examples of OTUB1 immunostaining in lung adenocarcinomas. Samples were scored as negative/low, medium, or high staining as described in Materials and Methods. Scale bar, 100 μm.

B   OTUB1 expression in lung non-small-cell carcinomas with different KRAS mutation status. OTUB1 expression was assessed in TMA samples by IHC.

C   OTUB1 up-regulation is an early event in lung adenocarcinoma development. Lung adenocarcinoma patients expressing different levels of OTUB1 were stratified by tumor stages (T1–T4). Total number of patients, n.

D   Representative examples of phosphorylated-ERK1/2 immunostaining in lung adenocarcinomas. Samples were scored as negative/low, medium, or high staining as described in Material and Methods. Scale bar, 100 μm.

E   OTUB1 levels correlates with increased levels of phosphorylated-ERK1/2 in wt KRAS lung adenocarcinomas. Association between OTUB1 and pERK1/2 expression levels was assessed in the same tumor samples by IHC.

F   Ki67 expression in lung adenocarcinomas expressing different levels of OTUB1. Ki67 expression was assessed by IHC. The percentage of Ki67-positive cells was counted, and the samples were scored as negative/low, < 10%; medium, between 10 and 40%; and high, > 40%. See Fig EV4, for representative images of Ki67 staining.

G   Wt KRAS lung adenocarcinoma TCGA patients with medium *OTUB1* expression have poorer overall survival.

H   *OTUB1* expression levels are not associated with survival rate of KRAS-mutant lung adenocarcinoma TCGA patients.

Data information: (G, H) Patients were stratified according to their OTUB1 mRNA levels and/or their KRAS status as described in Materials and Methods. Overall survival of lung adenocarcinoma patients expressing different levels of OTUB1 mRNA as measured by Kaplan–Meier curves. P-values were determined by log-rank test.

intermediate Ki67 staining (Figs 7F and EV4B). This could be explained by several observations that sustained hyperactivation of the MAPK pathway may result in growth inhibition rather than cell proliferation (Pumiglia & Decker, 1997; Ravi *et al*, 1998).

Multiple reports revealed that the expression of Ki67 in patients with stages T1–T3 NSCLC is a poor prognostic factor for survival (Martin *et al*, 2004). Consistently with these observations, we found that a moderate up-regulation of OTUB1 is significantly associated with poorer overall survival in TCGA lung adenocarcinoma patients with wt KRAS compared to low-expressing OTUB1 (Fig 7G). On the contrary, moderate OTUB1 expression levels do not affect the survival of patients harboring KRAS mutation (Fig 7H). Together, these results support that up-regulation of OTUB1 may play a crucial role in tumorigenic transformation of cells harboring wt KRAS by promoting RAS-induced ERK activation.

## Discussion

Our study identifies OTUB1 as a key player in the pathogenesis of wt KRAS lung cancers. Previously, OTUB1 has been implicated in the regulation of several physiological and pathological processes independently of its de-ubiquitinase activity. OTUB1 has been reported to regulate T-cell anergy by enhancing the degradation of the E3 ligase GRAIL (gene related to anergy in lymphocytes) (Lin *et al*, 2009), to augment TGF-β signaling by inhibiting degradation of phosphorylated SMAD2/3 (Herhaus *et al*, 2013), and to suppress DNA-damage-dependent chromatin ubiquitination (Nakada *et al*, 2010) and MDM2-mediated ubiquitination of the tumor suppressor p53 (Sun *et al*, 2012). Here we found that OTUB1 promotes tumorigenic transformation of wt RAS cells by triggering the RAS/MAPK pathway. Several lines of evidences support this conclusion. First, OTUB1 promotes hyperactivation of the MAPK cascade only when overexpressed in a wt RAS background. Second, OTUB1 replaces activated MEK1-mutant in an experimental model of human cell transformation. Furthermore, higher levels of OTUB1 expression are observed specifically in lung adenocarcinomas harboring wt KRAS compared to mutated RAS tumors. Finally, OTUB1 drives tumorigenic growth of wt RAS, but not mutant KRAS tumors.

OTUB1 triggers activation of the MAPK pathway by inhibiting RAS ubiquitination. Consistently with these observations, previous reports demonstrated that ubiquitination of HRAS and NRAS modulates their ability to activate the MAPK pathway (Xu *et al*, 2010; Yan *et al*, 2010). The RAS-specific E3 ligase RABEX5 may act as a tumor suppressor by regulating MAPK cascade activation. Rabex5-deficient mast cells exhibit enhanced and prolonged activation of Ras under basal conditions (Tam *et al*, 2004). A *Drosophila* Rabex5 hypomorphic mutation results in giant larvae or pupae, which often contain melanotic tumors (Yan *et al*, 2010). This phenotype is also attributed to dysregulation of RAS signaling (Zettervall *et al*, 2004; Yan *et al*, 2010).

In contrast to these studies and our results, some reports suggest that ubiquitination of KRAS at Lys117 and Lys147, which contribute to the formation of a GDP/GTP-binding pocket of RAS, increases its ability to activate downstream signaling and promote KRAS tumorigenic properties (Sasaki *et al*, 2011). However, the conclusions of these studies are mostly based on the analysis of Lys117 and Lys147-mutants (Baker *et al*, 2013a,b), whereas amino acid substitutions within the GDP/GTP-binding pocket have been shown to augment the flexibility of the pocket, leading to increased guanine nucleotide dissociation and higher GTP loading. Identification of E3 ubiquitin ligases that could ubiquitinate RAS proteins specifically at Lys117/Lys147 will help to resolve this apparent conundrum.

Whereas down-regulation or loss of function of RABEX5 in human cancers has not been reported, we found that OTUB1 expression is commonly up-regulated in a substantial subset of NSCLCs harboring wt KRAS. Recent studies also report increased expression of OTUB1 in prostate, colorectal, and breast cancers that are associated with poor survival, high metastatic potential, and chemotherapeutic drug resistance (Liu *et al*, 2014; Zhou *et al*, 2014; Iglesias-Gato *et al*, 2015; Karunarathna *et al*, 2015). Importantly, TCGA data analysis revealed that, similarly to lung adenocarcinomas, OTUB1 overexpression in colorectal cancer is mutually exclusive with KRAS mutations (co-occurrence log odds ratio: −3; *P*-value: 0.003), whereas RAS genes are rarely mutated in prostate and breast cancers (Karnoub & Weinberg, 2008). This further confirms that OTUB1 up-regulation may contribute to the development of different tumor types harboring wt RAS. Compellingly, a moderate up-regulation of OTUB1 decreases the overall survival of lung adenocarcinoma patients with wt KRAS, suggesting that only medium levels of OTUB1 confers advantage to cancer cells and/or could affect their chemotherapeutic resistance. On the other hand, the lack of effect of high OTUB1 expression levels on patients' survival could be further explained by the alternative interactions of OTUB1 beyond RAS that for instance were reported to lead to the stabilization of p53 (Sun *et al*, 2012) or CK2-mediated OTUB1 nuclear translocation to affect DNA repair (Herhaus *et al*, 2015).

Oncogenic mutations of the RAS GTPases have been observed in about one-third of human cancers (Karnoub & Weinberg, 2008). However, the high prevalence of RAS pathway activation strongly suggests the existence of alternative mechanisms of RAS/MAPK activation. Recently, RAS GAPs have emerged as an expanding new class of tumor suppressor genes that contribute to malignant transformation by triggering RAS activity (Maertens & Cichowski, 2014). Our data suggest that dysregulation of RAS ubiquitination represents an alternative mechanism to activate RAS during NSCLC development. Advancing knowledge on the regulatory hub controlling RAS ubiquitination and thus targeting RAS up-regulation could be clinically exploited as a strategy to inhibit the activation of wt RAS in lung cancers.

## Materials and Methods

### Cell culture and lentiviral transduction

HEK293T, HeLa, H838, H2009, H1437, A549, and A427 cells were cultured in DMEM-F12 medium (GIBCO); H1993, HOP62, and HOP92 were grown in RMPI 1640 medium (GIBCO). All media were supplemented with 10% fetal bovine serum and 1% penicillin/streptomycin. Embryonic kidney epithelial cells (HEK TEST) immortalized by hTERT, SV40 LT, and SV40 ST and expressing either empty vector, myristoylated AKT1 (myr-AKT), or MEK1 D218,

D222-mutant (MEKDD) were a gift from Dr. Hahn (Dana-Farber Cancer Institute, USA) (Boehm et al, 2007).

Transient transfections were performed using Turbofect, Lipofectamine LTX (Life Technologies), or XtremeGene9 (Roche). Lentiviral infections were performed as described by the RNAi Consortium (TRC). Infected cells were selected by treatment with 1–2 μg puromycin (InvivoGen) for 2 days.

## Expression vectors and antibodies

Full-length OTUB1 expression constructs were purchased from ORIGENE. Point mutations to generate OTUB1 C91S were obtained by QuickChange site-directed mutagenesis PCR (Stratagene, La Jolla). Lentiviral pLA-CMV-N-Flag or pLA-CMV-N-HA vectors were used to generate Flag/HA-tagged constructs. The pMT107–6×His–ubiquitin plasmid was a generous gift from Dr. Bohmann (University of Rochester, USA). pLKO.1-puro shGFP and pLKO.1-puro vectors containing shRNAs targeting OTUB1 (pLKO.1-shOTUB1_1 (TRCN0000004211), pLKO.1-shOTUB1_2 (TRCN0000004213), pLKO.1-shOTUB1_3 (TRCN0000004215)) were purchased from Sigma-Aldrich.

The antibodies used: mouse monoclonal anti-FLAG (Sigma-Aldrich, M2), anti-RAS (Millipore, Clone 10), anti-vinculin (Sigma-Aldrich, clone hVIN-1), anti-GAPDH (Sigma-Aldrich, GAPDH-71.1), anti-p44/42 MAPK (Erk1/2) (Thr202/Tyr204) (Cell Signaling, 3A7, #9107), anti-AKT (Cell Signaling, 40D4); rat monoclonal anti-HA (Roche, 3F10); rabbit monoclonal anti-phospho-p44/42 MAPK (Erk1/2) (Thr202/Tyr204) (IHC) (Cell Signaling, D13.14.4E, #4370), anti-Ki67 (Thermo Scientific #RM-9106-S, clone SP6); rabbit polyclonal anti-DYKDDDDK (Cell Signaling), anti-OTUB1 (Bethyl Laboratories), anti-OTUB1 (IHC) (Sigma-Aldrich, HPA039176), anti-phospho-p44/42 MAPK (Erk1/2) (Thr202/Tyr204) (Cell Signaling, #9101), anti-pAKT (Cell Signaling, D9E), anti-RABEX5 (Sigma-Aldrich).

## MAPPIT screen

MAPPIT experiments were performed as described previously (Lemmens et al, 2015). RAS proteins, expressed through a pSEL(+2L) vector, served as bait and DUB ORFs were cloned as prey in the pMG1 vector. The prey collection screened was selected from the human ORFeome collection versions 5.1 and 8.1 (http://horfdb.dfci.harvard.edu/hv5/). In brief, 293T cells were transfected with the bait and prey expressing plasmids combined with the STAT3-dependent pXP2d2-rPAPI-luciferase reporter plasmid. Twenty-four hours later, triplicate wells were supplemented with medium with or without 10 ng/ml erythropoietin. Twenty-four hours after leptin stimulation, luciferase activity was measured. MAPPIT signals were calculated as the ratio between the average values of the leptin-stimulated and the unstimulated samples.

## Immunoprecipitation, immunoblotting, and Meso Scale analysis

Cells were washed twice in cold PBS and scraped on ice in lysis buffer (50 mM Tris–HCl pH 7.5, 150 mM NaCl, 1% NP-40) containing protease inhibitor and phosphatase inhibitor cocktails (Roche). Samples were subsequently cleared by centrifugation for 10 min 16,000 g at 4°C. For immunoprecipitation assays, cells were lysed in co-immunoprecipitation buffer [50 mM Tris–HCl, at pH 7.5, 137 mM NaCl, 1% NP-40, 5 mM MgCl$_2$, 10% glycerol, and protease inhibitor cocktail (Roche)]. Tagged proteins were immunoprecipitated using anti-Flag (M2) or anti-HA agarose beads (Sigma-Aldrich) for 2 h at 4°C, washed five times with cold co-immunoprecipitation and finally eluted with 3 × Flag or HA peptides according to the manufacture's protocol. For immunoblotting, equivalent amounts of cell lysates were separated on 4–12% gradient gels (Invitrogen), transferred to nitrocellulose membranes, and incubated with the indicated antibodies. The signal was visualized with chemiluminescence detection reagent (Amersham Pharmacia Biotech) using an automated digital developer.

Meso Scale Discovery 96-well multispot plates were used for quantitative phospho/total ERK1/2(K15107D) and AKT1 (K15100A3) analyses according to the manufacturer's instructions (Meso Scale Diagnostics). Plates were analyzed using MESO Quick-Plex SQ120 multiplex imager (Meso Scale Diagnostics).

## RAS activation assay

The RAS activation assay was conducted according to the manufacturer's protocol (Millipore). Briefly, cells were washed with cold PBS and lysed in lysis/wash buffer (Millipore). Equal amounts of clarified cell lysates were mixed with Raf-1-RBD agarose beads (Millipore) and incubated for 45 min at 4°C. The beads were washed with lysis/wash buffer three times and eluted by boiling in 2 × SDS buffer. Eluted proteins were subjected to SDS–PAGE and immunoblotted.

## Purification of ubiquitinated proteins

HEK293T cells were co-transfected with 6×His–ubiquitin and Flag–RASs. Ubiquitinated proteins were purified as described previously (Simicek et al, 2013). Briefly, cells were lysed in co-immunoprecipitation buffer containing EDTA-free protease inhibitor cocktail (Roche). Cell lysates were mixed with His-buffer A (PBS, at pH 8.0, 6 M guanidinium–HCl, 0.1% NP-40 and 1 mM β-ME) and added to TALON beads (Clontech). After binding, the resin was washed with His-buffer B (PBS, at pH 8.0, 0.1% NP-40, 5% glycerol and 20 mM imidazole), and proteins were eluted in sample buffer.

For in vitro ubiquitination assay, purified proteins were incubated in the reaction buffer (20 mM Tris, pH 7.4; 50 mM NaCl; 100 μM ZnCl$_2$; 8 mM MgCl$_2$; 4 mM ATP) for 2 h at 30°C. The final protein concentrations in the reaction mix were as follows: UBE1 (80 nM), UbcH5C (400 nM), ubiquitin (16 μM), RABEX5 (1-76) (3 μM), Flag–NRAS (2.5 μM), and OTUB1 (3 μM). The reaction was quenched with 20 mM EDTA, and NRAS ubiquitination was analyzed by immunoblotting using anti-Flag M2 antibody.

## Fluorescence microscopy

For immunostaining, $2 \times 10^4$ HeLa cells were plated on 8-well chamber glass slide (Nalge Nunc International) and fixed 24 h after transfection with 4% PFA. Immunostaining was performed as described previously (Simicek et al, 2013). Briefly, cells were permeabilized in PBS-0.15% Triton and blocked with 1% BSA and 10% goat serum. Primary antibodies and goat Alexa-conjugated

secondary antibodies were applied by diluting in blocking buffer before mounting in Citifluor. Images were obtained by using a confocal Leica SPII microscope (63× magnification) (Leica Microsystems, Wetzlar, Germany).

Cellular distribution of RAS proteins was determined by automatic imaging using the IN Cell Analyzer 2000 system. Data analysis and image quantification were performed using ImageJ (National Institutes of Health USA).

### TCGA analysis

To generate the OncoPrint of OTUB1 gain and KRAS mutation status in lung adenocarcinoma (LUAD) and lung squamous cell carcinoma (LUSC), cBioPortal (http://www.cbioportal.org) (Cerami *et al*, 2012; Gao *et al*, 2013) was queried over all completed tumors from TCGA provisional datasets using the Onco Query Language (OQL), "OTUB1:GAIN; KRAS: MUT".

Additionally, clinico-pathological, gene expression, and copy number alteration data were downloaded from the TCGA data portal (https://tcga-data.nci.nih.gov/tcga/) for both datasets. For analysis of *OTUB1* expression, OTUB1 read counts from RNASeq analysis were normalized across all patient samples and log2-transformed. To determine the copy number status of OTUB1, the mean copy number segments overlapping the OTUB1 locus were extracted using a Perl script. OTUB1 gain was determined as samples with a log2 ratio > 0.1. Pearson correlation between *OTUB1* expression and copy number was performed and plotted in the R statistical package (Version 3.2.0). *Z*-score normalization of OTUB1 expression in tumors compared to matched normals revealed three levels of expression, negative/low (< 1 s.d.), medium (1–3 s.d.), and high (> 3 s.d.). Kaplan–Meier survival plots were generated in R comparing the overall survival for each level of OTUB1 expression in both KRAS wt and KRAS-mutated LUAD using the log-rank test.

### Immunohistochemistry

Non-small-cell lung cancer tissue micro-array slides (TristarGroup, US) were immunostained for OTUB1, Ki67, and pERKs on a Discovery Ventana automated staining platform. OTUB1 and phospho-ERKs immunohistochemistry was evaluated using a semi-quantitative approach that combines intensity and distribution of immunoreactivity in the epithelial tumor cells. Each single TMA spot was annotated using the same criteria applied in the HUMAN PROTEIN ATLAS project (http://www.proteinatlas.org/). Arrays were scored in a blinded manner on an intensity scale of 0–3 (0, no staining; 1, low staining; 2, moderate staining; 3, strong staining) and on a distribution scale of 0–3 (0, none; 1, < 25% tumor area; 2, between 25% and 75% tumor area; 3, > 75% tumor area). Data were then analyzed combining the intensity and distribution values in a scale of no/low staining (neg/low), moderate staining (medium), and strong staining (high). For Ki67 analysis, the percentage of positive cells was counted and the samples scored on a positivity scale as neg/low, < 10%; medium, between 10 and 40%; and high, > 40%.

### Anchorage-independent growth and tumor xenograft assays

Anchorage-independent growth in soft agar was performed as previously described in Boehm *et al* (2007). About $10^4$ cells were plated

in triplicates in 0.35% Noble agar over a 0.5% agar bottom layer. Three weeks after plating, several random areas were imaged and colonies were quantified using ImageJ software.

For tumor xenograft assays, $2.0 \times 10^6$ cells were injected subcutaneously into the lower flanks of 6-week-old female NMRI-nu (nu/nu) nude mice (Janvier). Tumor growth was monitored 3 times per week, and volumes were calculated using the following formula: volume = (tumor width$^2$ × tumor length)/2.

### Statistical analysis

Meso Scale measurements, RAS cellular distribution, and colony quantifications were calculated as percentages from at least three independent experiments. The error bars indicate the standard error of the mean (s.e.m.). *P*-values were calculated by two-tailed *t*-Test. All data were analyzed using GraphPad Prism for Apple Mac (version 6.0f). For the TCGA analysis, two-sided *t*-Tests determined the significant differences between groups means based on the expression levels. Two-way ANOVA was used to analyze xenograft experiments.

### Study approval

The ethical committee of the KU Leuven approved the animal study (declaration P166/2013).

**Expanded View** for this article is available online.

### Acknowledgements

We thank Dr. Tibor Pastor for helpful comments on the manuscript, and Jana Peeters and Magdalena De Troyer for technical support in generating stable cell lines and DNA constructs. This work was supported by the VIB (AS), Research Foundation Flanders (FWO) fellowships (MFB, MSi), FWO Research

### The paper explained

#### Problem
Hyperactivation of the RAS-MAPK oncogenic pathway is a common event in lung cancer. Despite many efforts to inhibit the RAS proteins, selective inhibition of RAS remains a considerable challenge; thus, a better understanding of the RAS pathway is urgently required to establish new treatment strategies.

#### Results
Recent reports demonstrate that reversible ubiquitination of RAS dramatically affects its activity, suggesting that enzymes involved in regulating RAS ubiquitination may contribute to malignant transformation. Our results strongly indicate that dysregulation by RAS mono-ubiquitination represents an alternative mechanism of RAS activation during lung cancer development. Specifically, we found that the de-ubiquitinase OTUB1 promotes lung cancer formation and correlates with poorer patient prognosis.

#### Impact
The development of small-molecule modulators of deubiquitinases has recently attracted the attention of the biomedical industry, rendering them promising targets for cancer treatment. The results of our study not only advance our understanding of RAS signaling, but also subsequently could lead to novel therapeutic approaches.

project G068715N (AS), ERC 105329 DELCANCER (AS), Stichting Tegen Kanker (F/2014/257) (AS), ERC 340941 CYRE (JT).

## Author contributions

MFB, MSi, LAA, MSt, VNA, and DMG performed the biochemical and cellular experiments; LAA did the xenografts; ER did the IHC staining and evaluation; SL and JT designed and performed the MAPPIT screen; JC did the bioinformatics analysis of TCGA data; MFB, MSi, and AAS analyzed the data; MFB and AAS wrote the manuscript. All authors discussed the results and commented on the manuscript.

## Conflict of interest

The authors declare that they have no conflict of interest.

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
