## [Review Process File · EMBO Molecular Medicine]

OTUB1 triggers lung cancer development by inhibiting RAS monoubiquitination

Maria Francesca Baietti, Michal Simicek, Layka Abbasi Asbagh, Enrico Radaelli, Sam Lievens, Mr. Jonathan Crowther, Mikhail Steklov, Vasily N Aushev, Martínez García, Jan Tavernier and Anna A Sablina

Corresponding author: Anna Sablina, VIB; KULeuven

Review timeline:

Submission date:	23 October 2015
Editorial Decision:	25 November 2015
Revision received:	01 December 2015
Editorial Decision:	18 December 2015
Accepted:	11 January 2016

Transaction Report:

Editor: Roberto Buccione

1st Editorial Decision

25 November 2015

Thank you for the submission of your manuscript to EMBO Molecular Medicine. We have now received comments from the Reviewers whom we asked to evaluate your manuscript.

As you will see the Reviewers are quite supportive and offer a few valuable suggestions to improve your manuscript and important requests for clarification.

I also agree that the manuscript would significantly benefit from some re-writing work to streamline the results and methods sections (without omitting the required information to ensure reproducibility of course) and to reduce redundancy.

In conclusion, while publication of the paper cannot be considered at this stage, we would be pleased to consider a suitably revised submission, provided that the Reviewers' concerns are fully addressed.

Please note that it is EMBO Molecular Medicine policy to allow a single round of revision only and that, therefore, acceptance or rejection of the manuscript will depend on the completeness of your responses included in the next, final version of the manuscript.

As you know, EMBO Molecular Medicine has a "scooping protection" policy, whereby similar findings that are published by others during review or revision are not a criterion for rejection. Although I do not foresee a delay in this case, I do ask you to get in touch with us after three months if you have not completed your revision, to update us on the status. Please also contact us as soon as possible if similar work is published elsewhere.

In the likely event of final acceptance, please also comply with the editorial requirements listed below:

1) As per our Author Guidelines, the description of all reported data that includes statistical testing must state the name of the statistical test used to generate error bars and P values, the number (n) of independent experiments underlying each data point (not replicate measures of one sample), and the actual P value for each test (not merely 'significant' or ' $P < 0.05$ ').

2) EMBO Molecular Medicine now requires a complete author checklist (<http://embomolmed.embopress.org/authorguide#editorial3>) to be submitted with all revised manuscripts. Provision of the author checklist is mandatory at revision stage; The checklist is designed to enhance and standardize reporting of key information in research papers and to support reanalysis and repetition of experiments by the community. The list covers key information for figure panels and captions and focuses on statistics, the reporting of reagents, animal models and human subject-derived data, as well as guidance to optimise data accessibility.

3) We are now encouraging the publication of source data, particularly for electrophoretic gels and blots, with the aim of making primary data more accessible and transparent to the reader. Would you be willing to provide a PDF file per figure that contains the original, uncropped and unprocessed scans of all or at least the key gels used in the manuscript? The PDF files should be labeled with the appropriate figure/panel number, and should have molecular weight markers; further annotation may be useful but is not essential. The PDF files will be published online with the article as supplementary "Source Data" files. If you have any questions regarding this just contact me.

4) Every published paper now includes a 'Synopsis' to further enhance discoverability. Synopses are displayed on the journal webpage and are freely accessible to all readers. They include a short standfirst as well as 2-5 one sentence bullet points that summarise the paper. Please provide the synopsis including the short list of bullet points that summarise the key NEW findings. The bullet points should be designed to be complementary to the abstract - i.e. not repeat the same text. We encourage inclusion of key acronyms and quantitative information. Please use the passive voice. Please attach this information in a separate file or send them by email, we will incorporate it accordingly. You are also welcome to suggest a striking image or visual abstract to illustrate your article. If you do please provide a jpeg file 550 px-wide x 400-px high.

5) Please comply with our new guidelines for presentation of supplementary data (<http://embomolmed.embopress.org/authorguide#expandedview>).

I look forward to receiving your revised manuscript as soon as possible.

***** Reviewer's comments *****

Referee #1 (Remarks):

This study aimed to demonstrate that the de-ubiquitinase OTUB1 is an inhibitor of RAS ubiquitination and can lead to the constraint of RAS on cell membrane, which is independent on its catalytic activity. In addition, the authors also indicated that the effect of OTUB1 in NSCLC with wild-type RAS is through the increased level of ERK1/2 phosphorylation. This manuscript provided a lot of data to support the relationship between OTUB1 and RAS in lung cancer progression, as well as the putative function or mechanism of OTUB1. Nevertheless, there are several drawbacks that undermine its suitability for publication in the current version and are listed below.

1. The legends of Fig 1E and 1F are not corresponding to the figures and should be reversed or corrected.
2. In Fig 2, the Western blotting data indicated that the experiments are pull-down assays, if so, it is not necessary to show "immunoprecipitates" and its abbreviation in figure legend.
3. In Figs 2A and 2D, the ubiquitinated NRAS and HRAS were purified by Co^{2+} metal affinity chromatography and detected by anti-Flag antibody. Perhaps misunderstand, this reviewer is wondering why the unmodified NRAS and HRAS (n.m. in the figures) can interact with His-tagged

ubiquitin?

4. The author indicated that stable knock-down of OTUB1 decreased anchorage-independent growth and suppressed the xenograft growth of the KRAS-mutant A549 cells (Figs 5C-E); however, the in vivo tumor growth of A549 was not affected while OTUB1 was overexpressed (Fig 6F). The authors should clarify this point.

5. The manuscript is too long and should be condensed, especially results and methodology sections. The authors can consider to delete the irrelevant or redundant description and wording and make the article more simplified.

Referee #2 (Remarks):

This study characterizes OTUB1 as a modulator of Ras activity during tumorigenic transformation. The major findings are 1) OTUB1 interacts with and deubiquitinates Ras proteins. 2) OTUB1-mediated deubiquitination increased Ras activity, which in turn activated the MAPK kinase pathway and enhanced tumorigenic transformation. On the other hand, the knock-down of OTUB1 decreased Ras activity and reduced the pathway activity and the tumorigenic potential of the cells. 3) The oncogenic function of OTUB1 was mediated only by wild type Ras, whereas it did not affect hyperactive mutant Ras proteins. 4) OTUB1 expression is associated in lung adenocarcinoma cells where K-RAS is wild type. The knock-down of OTUB1 decreased the growth of the lung cancer cells, while it did not affect the growth of the cells with mutant K-RAS. 5) Analysis of the sequencing and expression data of human cancers revealed the correlation of OTUB1 and wild type K-RAS status in non-small cell lung cancer samples, suggesting the physiological relevance of the protein interaction to the diseases.

These findings are novel and very interesting and may have potential impact in the cancer stratification and the development of novel therapeutic strategy. The experiments are well done with proper controls. The sets of gain- and loss-of-function experiments led to the sound conclusion in the manuscript. However, a main question has been raised about the method of the analysis that led to the data in Figure 4 and Figure 7G and H. For example, an analysis of the same data set using the cBioportal do not produce the data in Figure 4A. Also, it is unclear how the rest of the analysis data was generated. This concern can be addressed with some explanation and a user-friendly, detailed guideline to repeat the analysis to get to the same findings.

Minor points are a) Figure 1B needs a little more description about REM2 and EFHA1 in both in the main text and the figure legend; b) A little more discussion is needed to explain or speculate why A549 with K-RASG12S is still inhibited by OTUB1 knock-down, which will better explain the phenotype associated with the knock-down; c) Any discussion as to why AKT is not affected by OTUB1 will be helpful.

Referee #3 (Remarks):

The work by Baietti et al. delivers exciting and important new information decoding the mechanistic basis of dynamic post-translational features that specify adaptive and maladaptive activity of Ras proteins. It has significance both with respect to generalizable principles of signal transduction mechanisms and with respect to specific mechanisms supporting Ras-induced tumorigenesis. The work is rigorously performed, convincing and effectively presented. I expect the paper to be well received and highly cited. I have no substantive concerns about the scientific content, biological relevance or significance of the advance.

Reviewer's comments

Referee #1 (Remarks):

This study aimed to demonstrate that the de-ubiquitinase OTUB1 is an inhibitor of RAS ubiquitination and can lead to the constraint of RAS on cell membrane, which is independent on its catalytic activity. In addition, the authors also indicated that the effect of OTUB1 in NSCLC with wild-type RAS is through the increased level of ERK1/2 phosphorylation. This manuscript provided a lot of data to support the relationship between OTUB1 and RAS in lung cancer progression, as well as the putative function or mechanism of OTUB1. Nevertheless, there are several drawbacks that undermine its suitability for publication in the current version and are listed below.

1. The legends of Fig 1E and 1F are not corresponding to the figures and should be reversed or corrected.
2. In Fig 2, the Western blotting data indicated that the experiments are pull-down assays, if so, it is not necessary to show "immunoprecipitates" and its abbreviation in figure legend.
3. In Figs 2A and 2D, the ubiquitinated NRAS and HRAS were purified by Co²⁺ metal affinity chromatography and detected by anti-Flag antibody. Perhaps misunderstand, this reviewer is wondering why the unmodified NRAS and HRAS (n.m. in the figures) can interact with His-tagged ubiquitin?
4. The author indicated that stable knock-down of OTUB1 decreased anchorage-independent growth and suppressed the xenograft growth of the KRAS-mutant A549 cells (Figs 5C-E); however, the in vivo tumor growth of A549 was not affected while OTUB1 was overexpressed (Fig 6F). The authors should clarify this point.
5. The manuscript is too long and should be condensed, especially results and methodology sections. The authors can consider to delete the irrelevant or redundant description and wording and make the article more simplified.

Referee #2 (Remarks):

This study characterizes OTUB1 as a modulator of Ras activity during tumorigenic transformation. The major findings are 1) OTUB1 interacts with and deubiquitinates Ras proteins. 2) OTUB1-mediated deubiquitination increased Ras activity, which in turn activated the MAPK kinase pathway and enhanced tumorigenic transformation. On the other hand, the knock-down of OTUB1 decreased Ras activity and reduced the pathway activity and the tumorigenic potential of the cells. 3) The oncogenic function of OTUB1 was mediated only by wild type Ras, whereas it did not affect hyperactive mutant Ras proteins. 4) OTUB1 expression is associated in lung adenocarcinoma cells where K-RAS is wild type. The knock-down of OTUB1 decreased the growth of the lung cancer cells, while it did not affect the growth of the cells with mutant K-RAS. 5) Analysis of the sequencing and expression data of human cancers revealed the correlation of OTUB1 and wild type K-RAS status in non-small cell lung cancer samples, suggesting the physiological relevance of the protein interaction to the diseases.

These findings are novel and very interesting and may have potential impact in the cancer stratification and the development of novel therapeutic strategy. The experiments are well done with proper controls. The sets of gain- and loss-of-function experiments led to the sound conclusion in the manuscript. However, a main question has been raised about the method of the analysis that led to the data in Figure 4 and Figure 7G and H. For example, an analysis of the same data set using the cBioportal do not produce the data in Figure 4A. Also, it is unclear how the rest of the analysis data

was generated. This concern can be addressed with some explanation and a user-friendly, detailed guideline to repeat the analysis to get to the same findings.

Minor points are a) Figure 1B needs a little more description about REM2 and EFHA1 in both in the main text and the figure legend; b) A little more discussion is needed to explain or speculate why A549 with K-RASG12S is still inhibited by OTUB1 knock-down, which will better explain the phenotype associated with the knock-down; C) Any discussion as to why AKT is not affected by OTUB1 will be helpful.

Referee #3 (Remarks):

The work by Baietti et al. delivers exciting and important new information decoding the mechanistic basis of dynamic post-translational features that specify adaptive and maladaptive activity of Ras proteins. It has significance both with respect to generalizable principles of signal transduction mechanisms and with respect to specific mechanisms supporting Ras-induced tumorigenesis. The work is rigorously performed, convincing and effectively presented. I expect the paper to be well received and highly cited. I have no substantive concerns about the scientific content, biological relevance or significance of the advance.

Detailed Response to reviewers

Referee #1

1. The legends of Fig 1E and 1F are not corresponding to the figures and should be reversed or corrected.
2. In Fig 2, the Western blotting data indicated that the experiments are pull-down assays, if so, it is not necessary to show "immunoprecipitates" and its abbreviation in figure legend.

We thank the Reviewer for identifying these errors, which we have corrected in the revised manuscript.

3. In Figs 2A and 2D, the ubiquitinated NRAS and HRAS were purified by Co²⁺ metal affinity chromatography and detected by anti-Flag antibody. Perhaps misunderstand, this reviewer is wondering why the unmodified NRAS and HRAS (n.m. in the figures) can interact with His-tagged ubiquitin?

In some of the cases, there is non-specific protein binding to the surface of Talon beads (crosslinked agarose loaded with Co²⁺) that is difficult to avoid even with more stringent washing conditions.

4. The author indicated that stable knock-down of OTUB1 decreased anchorage-independent growth and suppressed the xenograft growth of the KRAS-mutant A549 cells (Figs 5C-E); however, the in vivo tumor growth of A549 was not affected while OTUB1 was overexpressed (Fig 6F). The authors should clarify this point.

The RAS pathway is already maximally active in A549 cell due to the constitutively-active RAS mutation, and an increase of OTUB1 expression did not lead to additional hyperactivation of the RAS signaling in these cells (Figure 6B,F). It could explain why OTUB1 did not facilitate tumor growth in A549 cells.

On the other hand, suppression of OTUB1 in A549 cells led to down-regulation of the RAS signaling pathway and inhibition of tumorigenic growth (Figure 5), indicating that OTUB1 is

essential to maintain the activity of KRAS-G12V. We have clarified this issue in the revised manuscript.

5. The manuscript is too long and should be condensed, especially results and methodology sections. The authors can consider to delete the irrelevant or redundant description and wording and make the article more simplified.

We thank the reviewer for this comment; we have considerably shortened the revised manuscript.

Referee #2

The sets of gain- and loss-of-function experiments led to the sound conclusion in the manuscript. However, a main question has been raised about the method of the analysis that led to the data in Figure 4 and Figure 7G and H. For example, an analysis of the same data set using the cBioportal do not produce the data in Figure 4A. Also, it is unclear how the rest of the analysis data was generated. This concern can be addressed with some explanation and a user-friendly, detailed guideline to repeat the analysis to get to the same findings.

We thank the reviewer for highlighting these issues. To generate the oncoprint of OTUB1 gain and KRAS mutation status in lung adenocarcinoma (LUAD) and lung squamous cell carcinoma (LUSC) from TCGA provisional datasets (Figure 4a), Cbioportal (Cerami et al, 2012; Gao et al, 2013) was queried using the Onco Query Language (OQL), OTUB1:GAIN;KRAS:MUT;.

To perform all other TCGA data analyses, we downloaded clinicopathological, gene expression, and copy number alteration data from the TCGA data portal, <https://tcga-data.nci.nih.gov/tcga/>, for both datasets. We have clarified how we performed the analyses in the revised Materials and methods and figure legends.

Minor points are

- a) Figure 1B needs a little more description about REM2 and EFHA1 in both in the main text and the figure legend;

REM2 and EFHA1 were identified as recurrent hits in multiple MAPPIT screens. We found that either REM2, or EFHA1 triggers a strong MAPPIT signal regardless of which bait protein is fused to the MAPPIT receptor, indicating that the REM2 and EFHA1 proteins bind to the MAPPIT bait receptor. Here we used REM2 and EFHA1 to control functional expression of the MAPPIT bait receptors. We have included this explanation to the revised Figure legend 1.

- b) A little more discussion is needed to explain or speculate why A549 with K-RASG12S is still inhibited by OTUB1 knock-down, which will better explain the phenotype associated with the knock-down;

Please see the reply to reviewer #1, point 4.

- c) Any discussion as to why AKT is not affected by OTUB1 will be helpful.

The reviewer is correct that we did not observe AKT activation upon OTUB1 overexpression. This could be due to OTUB1-mediated inhibition of TRAF6 (Li et al., JBC, 2010), which is

essential for AKT activation (Yang et al., Science, 2009). We have provided this explanation in the revised result section.

2nd Editorial Decision

18 December 2015

Thank you for the submission of your revised manuscript to EMBO Molecular Medicine. We have now received the enclosed reports from the referees that were asked to re-assess it. As you will see the reviewers are now globally supportive and I am pleased to inform you that we will be able to accept your manuscript pending the following final amendments:

1) Although I had asked you to provide the Author checklist (<http://embomolmed.embopress.org/authorguide#editorial3>) in my previous decision letter, you have not done so. Please include it in the next revision.

2) I had also asked you to provide a synopsis for your article: "Synopses are displayed on the journal webpage and are freely accessible to all readers. They include a short standfirst as well as 2-5 one sentence bullet points that summarise the paper. Please provide the synopsis including the short list of bullet points that summarise the key NEW findings. The bullet points should be designed to be complementary to the abstract - i.e. not repeat the same text. We encourage inclusion of key acronyms and quantitative information. Please use the passive voice. Please attach this information in a separate file or send them by email, we will incorporate it accordingly. You are also welcome to suggest a striking image or visual abstract to illustrate your article. If you do please provide a jpeg file 550 px-wide x 400-px high."

3) Please provide two additional keywords and a conflict of interest statement.

4) Thank you for providing source data. I have noted a few issues that require your attention:

- a) The FLAG band in the 2nd strip of Fig. 1F does not appear to match the corresponding source data
- b) Please check if labelling of loading controls for Fig. 2 A, B, C is correct in the source data file (is one meant to be GAPDH?).
- c) There is a missing label on one strip in the source data for Fig 2D (vimentin?).
- d) Shouldn't the label for figure "S6" actually be S3?
- d) Please provide one separate pdf source data file (multi page is fine) per each figure. Essentially you just need to split into one pdf per figure.

5) Please correct the callouts in the manuscript and the figure labels for the appendix data (should be Appendix figure S1, S2.... etc; <http://embomolmed.embopress.org/authorguide#expandedview>). Please also note that this correction should also be performed for the source data files.

6) Please provide scale bars for Fig. 7A and EV4.

Please submit your revised manuscript within two weeks. I look forward to seeing a revised form of your manuscript as soon as possible.

***** Reviewer's comments *****

Referee #1 (Remarks):

The authors have addressed the concerns. This reviewer has no further comments.

Referee #2 (Remarks):

The authors responded to the critiques very well. Although their revised method for analyzing TCGA data, especially correlation of OTUB1 and KRAS mutations status or patient outcomes, is

not completely user-friendly yet, it is appropriate given the limited space for the method section. Overall this revised manuscript is suitable for publication.